Dynamic effects of irrigation on photosynthesis and yield-related physiological characteristics in different glutinous wheat cultivars

Li Yan lylxy0524@126.com 1
Wang Xin 1
Kang Ruoxi 2
Xu Lixiao 1
Li Xuegui 1
Liu Hanyu 1
Qiu Zhennan 1
Dai Zhongmin 1
Zhu Yuangang zhuyuangang2002@163.com 1
1 College of Life Science, Dezhou University , Dezhou , Shandong , China
2 College of Life Science, Chongqing Normal University , Chongqing , China
Kutlu Imren
Electronic publication date: 2025 Oct 24
Publication date: 2025
Volume: 13
Electronic Location ID: e20230
Received 2025 May 6; Accepted 2025 Sep 23
Copyright: ©2025 Li et al.
Copyright year: 2025
Copyright holder: Li et al.
License: This is an open access article distributed under the terms of the Creative Commons Attribution License, which permits unrestricted use, distribution, reproduction and adaptation in any medium and for any purpose provided that it is properly attributed. For attribution, the original author(s), title, publication source (PeerJ) and either DOI or URL of the article must be cited.
License URL: https://creativecommons.org/licenses/by/4.0/

Keywords: Irrigation, Glutinous wheat, Photosynthesis, Antioxidant enzymes, Starch synthases

Funding: The National Natural Science Foundation of China No. 31271667 The Shandong Provincial Natural Science Foundation No. ZR2019MC032 The Open Project Program of State Key Laboratory for Crop Stress Resistance and High-Efficiency Production NWAFU SKLCSRHPKF14 This work was supported by the National Natural Science Foundation of China (No. 31271667), the Shandong Provincial Natural Science Foundation (No. ZR2019MC032) and the Open Project Program of State Key Laboratory for Crop Stress Resistance and High-Efficiency Production NWAFU (SKLCSRHPKF14). The funders had no role in study design, data collection and analysis, decision to publish, or preparation of the manuscript.

==============================
Water scarcity critically constrains wheat production in North China, yet the irrigation responsiveness of novel glutinous wheat cultivars remains poorly quantified. This study systematically investigated the physiological mechanisms of common wheat Shimai 19 (SM19), partially glutinous SM19-P (Wx-B1 null), and fully glutinous SM19-N (triple null) under three irrigation regimes: rain-fed (W0), water-saving (W1: jointing irrigation), and conventional irrigation (W2: overwintering + jointing + flowering irrigations). Dynamic monitoring of flag leaf photosynthesis (Soil Plant Analysis Development (SPAD), stomatal conductance (Gs), transpiration rate (Tr), net photosynthetic rate (Pn)), antioxidant enzyme systems (superoxide dismutase (SOD), peroxidase (POD), catalase (CAT) net photosynthetic rate (Pn), malondialdehyde (MDA)), grain starch synthase activities (granule-bound starch synthase (GBSS), soluble starch synthase (SSS), adenosine diphosphate glucose pyrophosphorylase (AGPase), starch branching enzyme (SBE)), and yield components revealed that: (1) SM19 achieved maximum photosynthetic capacity under W1 (Pn increased by 59.54% vs. W0 at 21 days post anthesis (DPA), p < 0.05) with optimal oxidative damage resistance (MDA reduced by 78.73% at 0 DPA), whereas SM19-P and SM19-N required W2 to reach photosynthetic peaks (Pn increased by 60.56% in SM19-P) and enzyme activity maxima (CAT increased by 66.67% in SM19-N). (2) Starch synthase activities peaked under water deficit (W0) early in grain-filling (≤14 DPA) but became tightly coupled to irrigation frequency thereafter. This was supported by a highly significant correlation between irrigation and final yield (r = 0.803, p < 0.01). The coordinated upregulation of AGPase and SSS (r = 0.726, p < 0.01) underpinned this response. The superior branched-starch accumulation in genotype SM19-N (+23%) was linked to its markedly higher SBE activity (r = 0.867, p < 0.01). (3) Yield optimization was genotype-specific: SM19 yielded highest under W1, while SM19-N peaked under W2. The study demonstrates that, unlike common wheat (SM19) which performs optimally under water-saving irrigation, the novel glutinous lines (SM19-P/SM19-N) require full irrigation to realize their yield potential, highlighting a critical trade-off between starch quality and drought adaptation. The key indicators identified—photosynthetic efficiency, antioxidant capacity, and starch metabolism—provide a theoretical foundation for developing future glutinous wheat varieties combining drought tolerance with high starch quality.

Introduction

Wheat (Triticum aestivum L.) is one of China’s three major food crops, with the wheat planting area accounting for about 20% of the total cultivated land in China. Water scarcity has always been a major limiting factor for agricultural production in China’s main wheat-producing areas (Ma et al., 2025). In arid and semi-arid regions such as North China, the combined impact of rising temperatures and reduced precipitation has led to an increase in the water deficit and intensified drought, ultimately resulting in a decrease in the wheat yield and planting area (Zeng et al., 2023). The analysis of data from 44 meteorological stations in the North China Plain from 1981 to 2017 revealed that drought during four growth stages of winter wheat (sowing-overwintering, return green-jointing, jointing-flowering, and flowering-maturity) had the most significant impact on yield, with drought during the flowering-maturity stage causing the most obvious reduction in winter wheat yield at a reduction rate of over 26.8% in Shandong Province (Sun et al., 2021). Improving the quality and yield of wheat in this region and achieving efficient utilization of water resources are urgent issues to be solved for the development of sustainable agricultural in China.

Physiological and morphological characteristics have significant effects on the growth and development of wheat, and enhancing wheat photosynthesis is considered an important strategy for increasing wheat yield (Li et al., 2023). Under different water conditions, the chlorophyll content of wheat flag leaves varies. The chlorophyll content is significantly positively correlated with wheat yield, with a higher correlation under irrigation conditions than under drought stress (Ahsan et al., 2022). Muhammad et al. (2018) subjected two winter wheat cultivars to moderate and severe drought stress at the tillering and jointing stages, followed by re-watering, and systematically measured photosynthetic rate, membrane stability, reactive oxygen species (ROS) content, antioxidant enzyme (SOD, CAT, superoxide dismutase (SOD), peroxidase (POD), ascorbate peroxidase (APX)) activities, osmolyte concentrations and yield components. Their results indicate that net photosynthetic rate (Pn), malondialdehyde (MDA) content, catalase (CAT) activity and superoxide dismutase (SOD) activity are four key physiological indicators for assessing drought resistance in wheat. Studies have shown that the photosynthetic capacity in different wheat varieties are significantly positively correlated with the wheat yield (Mahdavi et al., 2021). Starch is an important carbohydrate in wheat grains, and the starch accumulation directly affects the grain yield (Huang et al., 2021). The synthesis of starch is regulated by a series of enzymes (Ran et al., 2020). Adenosine diphosphate glucose pyrophosphorylase (AGPase, EC2.7.7.27), soluble starch synthase (SSS, EC2.4.1.21), granule-bound starch synthase (GBSS, EC2.4.1.21), and starch branching enzyme (SBE, EC2.4.1.18) play important roles in starch synthesis (Okpala et al., 2022). The accumulation rates of straight-chain and branched-chain starch in wheat are directly related to the activities of SBE, SSS, GBSS, and AGPase (Wang et al., 2014). The activity of starch synthesis-related enzymes in grains is sensitive to water (Xin et al., 2023). A water deficit during the grain-filling period inhibits the activities of key enzymes in wheat starch synthesis (Dai, Yin & Wang, 2009). However, some studies suggest that a water deficit is beneficial for increasing the activity of starch synthesis-related enzymes in wheat during the early and middle stages of grain filling, while the activity of starch synthesis enzymes significantly decreases in the later stage (Abid et al., 2018; Abdoli & Saeidi, 2013; Lv et al., 2021; Singh et al., 2008; Xin et al., 2023).

The partial or complete absence of enzymes controlling the synthesis of straight-chain starch in glutinous wheat results in increased amylopectin content, with unique starch characteristics and nutritional quality, conferring glutinous wheat with great value and application prospects. Since the first glutinous wheat variety was bred in China by You (2000), research on glutinous wheat has mainly focused on aspects such as the processing quality, starch structure characteristics, and storage starch synthesis mechanisms of glutinous wheat. For example, Giannou & Tzia (2016) investigated the quality of different glutinous wheat varieties and the characteristics of their baked products to promote the quality improvement and development of glutinous wheat. In addition, Guzmán et al. (2012) characterized novel Waxy (wx) alleles in spelt wheat, providing valuable genetic resources for the breeding of waxy wheat with altered starch properties. Graybosch (1998) systematically characterized the natural GBSS mutations that produce “partial waxy” and “full waxy” wheats, noting that their starch properties resemble those of waxy maize and hold broad promise for applications in Asian wet noodles, modified food starches, bakery shelf-life extension, and gluten co-production. Dai, Liu & Qin (2021) examined the impact of different irrigation methods on the starch components and physicochemical properties of glutinous wheat. At present, there are reports on the effects of combined irrigation on common winter wheat, but there is relatively little research on the post-flowering yield-related physiological mechanisms of glutinous wheat under different irrigation conditions. In the present study, common wheat and its derived partially glutinous and fully glutinous varieties were utilized to explore the dynamic changes in flag leaf photosynthetic characteristics, antioxidant enzyme and starch synthesis enzyme activities, and wheat yield during four post-flowering periods under three irrigation conditions. Unlike previous studies focusing solely on common wheat varieties, our study highlights the response of partially and fully glutinous wheat varieties (SM19-P, SM19-N) under varying irrigation frequencies. This work provides a theoretical basis and practical foundation for the further breeding of drought-resistant and high-yielding glutinous wheat varieties in the North China Plain.

Materials & Methods

Experimental materials

This study utilized the certified drought-resistant cultivar Shimai 19 (JiShenMai 2003001) as the genetic background to develop novel glutinous wheat lines SM19-P (Wx-B1 missing) and SM19-N (Wx-A1, Wx-B1, and Wx-D1 all missing) through hybrid backcross breeding with the fully glutinous germplasm Tiannuo 158. These lines exhibit significantly reduced amylose content. Although starch traits remain stable in the new lines (three-year coefficient of variation <3%), the physiological mechanisms underlying waxy modification’s impact on water stress response remain unquantified. Through controlled irrigation experiments, we systematically evaluated water use efficiency, and grain yield stability of SM19-P and SM19-N under drought conditions, providing data support for cultivar deployment in the Huang-Huai dryland wheat region.

Experimental design

The experiment was conducted from 2022 to 2023 at the Dezhou Science and Technology Park Experimental Farm (37°N, 116°E) in Shandong Province, China. The soil was loam soil, and the previous crop was maize. The topsoil (0–20 cm) characteristics were: pH 8.13, organic matter 10.65 g kg−1, total nitrogen 1.34 g kg−1, available phosphorus 22.56 mg kg−1, available potassium 140.9 mg kg−1. Three irrigation cultivation conditions (Table 1) were established. Each experimental plot had an area of 3 m × 4 m = 12 m2 and was arranged in a randomized complete block design with three replications. Baseline fertilization (120 kg N hm−2, 120 kg P2O5 hm−2, 120 kg K2O hm−2) was applied throughout the trial. Other field management measures were conducted conventionally. The 2022–2023 wheat season received 180.6 mm of precipitation, 15% below average, marked by a severe winter drought (Dec–Feb: 26.7 mm, 49% deficit) but above-average rainfall during critical spring growth stages (April: +30%; May: +7%). Temperatures were notably cold in winter (Jan: −2.8 °C) before rising to a warm June (26.4 °C). Moisture availability was uneven, with winter moisture stress followed by adequate spring supply during jointing and grain filling.

Measurement items and methods

Wheat plants flowering on the same day were marked with tags in preparation for the measurement of various parameters.

Measurement of photosynthetic parameters

Pn represents the leaf CO2 assimilation capacity. On clear days between 09:30 and 10:30 (10:00 ± 30 min), five healthy flowering plants were randomly selected per plot. Using the Li-6400 system (light intensity: 1,500 µmol m−2 s−1, CO2: 400 ppm, flow rate: 500 µmol s−1), measurements of Pn, Gs, and Tr were taken on the flag leaves of the main stems. A single leaf was measured per plant, with the mean value of five plants representing the plot.

Measurement of leaf SPAD values

To accurately capture the physiological responses of crops under high-temperature conditions, SPAD measurements were taken at noon in this study to maximize the exposure of differences among drought-resistant genotypes, the SPAD value of tagged wheat flag leaves was determined using a handheld chlorophyll meter (SPAD-502; Minolta, Tokyo, Japan). The average value of 10 leaves was taken as the measurement value. Because the SPAD-502 reading was closely related to the chlorophyll content (Alison et al., 2020), it directly represented the chlorophyll content.

Measurement of antioxidant enzyme activity

Leaf samples weighing 0.2 g were added to five mL of enzyme extraction solution (50 mmol L−1 pH 7.8 phosphate buffer, 0.3% triton-100, 4% polyvinylpyrrolidone, and 0.1 mmol EDTA L−1). After grinding in an ice bath, the mixture was transferred into a centrifuge tube. The tube was centrifuged at 12,000 ×g for 20 min under cold conditions, and the supernatant was collected for further use. The malondialdehyde (MDA) content was determined using the thiobarbituric acid method (Mammadhasanova, Sultanova & Fataliyev, 2023). The SOD activity was obtained following the nitro blue tetrazolium method, with 50% inhibition of nitro blue tetrazolium photochemical reduction defined as one unit of enzyme activity (Giannopolitis & Ries, 1977). The CAT activity was determined using ultraviolet spectrophotometry, with the enzyme activity expressed as the absorbance value of H2O2 decomposition per gram of fresh weight per minute (Teranishi et al., 1974). The guaiacol colorimetric method was employed to obtain the POD activity, with one unit of enzyme activity defined as a decrease in the POD value of 0.01 per minute (Weng et al., 2015).

Table 1 Irrigation experiment design.

Treatment	Irrigation amount during each growth stage (m3 hm−2)	Number of irrigations	Total irrigation amount (m3 hm−2)	
	Wintering	Jointing	Flowering			
W0 (rainfed)	–	–	–	0	0	
W1 (water-saving irrigation)	–	750	–	1	750	
W2 (conventional irrigation)	750	750	750	3	2,250	

Measurement of amylase activity

Ten wheat plants were selected from each plot at 7, 14, 21, 28, and 35 days after flowering. The first and second grains in the middle of the spike were marked and stored at −80 °C for the physiological measurement of enzyme activity. The activities of GBSS, SSS, AGPase, and SBE were determined using an amylase assay kit (Jingmei Biotechnology Co., Ltd., Suzhou, China). The GBSS, SSS, and AGPase activities were expressed as the amount of nicotinamide adenine dinucleotide phosphate—reduced form produced per gram of fresh plant weight per minute (nmol), while the SBE activity was expressed as the percentage decrease in absorbance at a wavelength of 660 nm. Each 1% decrease in iodine blue value per gram of plant fresh weight in the reaction system was considered as one enzyme activity unit.

Measurement of yield-related indicators

Wheat was sown on October 6, 2022, with a basic seedling density of 120,000 per 667 m2. It was harvested by plot on June 8, 2023, and the yield per unit area was calculated. and the spike number per unit area, grain number per spike, thousand-grain weight were determined simultaneously.

Data analysis

Data organization and preliminary calculations were performed using Microsoft Excel 2010. All data were subjected to standard analysis of variance (ANOVA) and correlation analysis using SPSS 21.0. Means and significant differences between treatments were separated using the least significant differences (LSD) test at the 5% probability level.

Results

Effects of different irrigation treatments on wheat photosynthetic parameters

Figure 1 illustrates dynamic changes in photosynthetic characteristics of wheat flag leaves under different irrigation treatments. Across varieties, photosynthetic parameters followed consistent developmental trends: net photosynthetic rate (Pn) initially increased then declined significantly, with SM19-P showing the greatest reduction under rain-fed conditions (W0). SM19 achieved peak Pn under water-saving irrigation (W1), exhibiting significant increases compared to W0 (e.g., 59.54% higher at 21 days after anthesis, days after anthesis (DAA); p < 0.05). Both SM19-P and SM19-N maximized Pn under conventional irrigation (W2), with SM19-P showing the largest improvement (60.56% increase vs. W0 at 21 DAA; p < 0.05). Trends in stomatal conductance (Gs), transpiration rate (Tr), and SPAD mirrored Pn patterns. These results demonstrate that SM19 benefits most from single irrigation (W1) under adequate post-flowering rainfall, while SM19-P and SM19-N require full irrigation (W2) for optimal photosynthesis due to higher water requirements.

Figure 1 Effects of different irrigation treatments on the photosynthetic parameters of wheat.

Vertical bars represent ± standard deviation (SD) of the mean (n = 3); different letters on the SD bars indicate significant differences (p < 0.05). SPAD, relative chlorophyll content; Pn, net photosynthetic rate; Gs, stomatal conductance; Tr, transpiration rate; DAA, days after anthesis; W0, rain-fed conditions; W1, water-saving irrigation with one irrigation event; W2, conventional irrigation with three irrigation events. (A) SM19, SPAD; (B) SM19-P, SPAD; (C) SM19-N, SPAD; (D) SM19, Pn; (E) SM19-P, Pn; (F) SM19-N, Pn; (G) SM19, Gs; (H) SM19-P, Gs; (I) SM19-N, Gs; (J) SM19, Tr; (K) SM19-P, Tr; (L) SM19-N, Tr.

Effects of different irrigation treatments on the antioxidant enzyme activity of wheat

Figure 2 depicts dynamic changes in antioxidant enzyme activities of wheat flag leaves. Across varieties, SOD activity progressively increased until 21 days after flowering (DAA), while POD and CAT activities peaked earlier at 14 DAA and 7 DAA, respectively. SM19 exhibited maximal SOD, POD, and CAT activities under water-saving irrigation (W1), with all enzymes showing significantly higher values versus rain-fed conditions (W0) throughout development (p < 0.05). The most pronounced enhancement occurred in CAT activity under W1, reaching 160.52% above W0 levels at 21 DAA (Fig. 2G). Minimal differences were observed between W1 and conventional irrigation (W2) for all enzymes.

Figure 2 Effects of different irrigation treatments on the antioxidant enzyme activity of wheat.

Vertical bars represent ± standard deviation (SD) of the mean (n = 3); different letters on the SD bars indicate significant differences (p < 0.05). SOD, superoxide dismutase; POD, peroxidase; CAT, catalase; DAA, days after anthesis; W0, rain-fed conditions; W1, water-saving irrigation with one irrigation event; W2, conventional irrigation with three irrigation events. (A) SM19, SOD; (B) SM19-P, SOD; (C) SM19-N, SOD; (D) SM19, POD; (E) SM19-P, POD; (F) SM19-N, POD; (G) SM19, CAT; (H) SM19-P, CAT; (I) SM19-N, CAT.

Both SM19-P and SM19-N exhibited increased SOD, POD, and CAT activities with higher irrigation frequency (Figs. 2B, 2C, 2E, 2F, 2H, 2I). Under conventional irrigation (W2), all enzymes showed significant enhancements versus rain-fed conditions (W0) (p < 0.05), with CAT activity demonstrating the most substantial improvements (up to 66.67% increase in SM19-N at 0 DAA; Figs. 2H, 2I).

MDA content increased temporally across varieties, peaking at 21 DAA (Fig. 3). SM19 achieved minimal MDA under water-saving irrigation (W1), showing significant reductions versus W0 (e.g., 78.73% lower at 0 DAA; p < 0.05; Fig. 3A). Conversely, SM19-P and SM19-N required conventional irrigation (W2) for optimal MDA reduction, with SM19-P exhibiting the largest decrease (64.80% at 0 DAA; p < 0.05; Fig. 3B).

Figure 3 Effects of different irrigation treatments on wheat malondialdehyde (MDA) content.

Vertical bars represent ± standard deviation (SD) of the mean (n = 3); different letters on the SD bars indicate significant differences (p < 0.05). DAA, days after anthesis; W0, rain-fed conditions; W1, water-saving irrigation with one irrigation event; W2, conventional irrigation with three irrigation events. (A) SM19; (B) SM19-P; (C) SM19-N.

Effects of different irrigation treatments on wheat starch synthase activity

The GBSS activity of SM19 and SM19-P increased first and then decreased with time, and the GBSS activity of SM19 was higher than that of SM19-P under the same conditions. The GBSS activity of SM19 first increased and then decreased with the increase in irrigation frequency before 14 days after anthesis; it first increased and then decreased after 21 days, with the highest activity under water-saving conditions (Fig. 4A). The GBSS activity of SM19-P decreased with the increase in irrigation frequency before 14 days after anthesis and then increased with the increase in irrigation frequency after 21 days (Fig. 4B).

Figure 4 Effect of irrigation on the granule-bound starch synthase (GBSS) activity of wheat.

Vertical bars represent ± standard deviation (SD) of the mean (n = 3); different letters on the SD bars indicate significant differences (p < 0.05). DAA, days after anthesis; W0, rain-fed conditions; W1, water-saving irrigation with one irrigation event; W2, conventional irrigation with three irrigation events. (A) SM19; (B) SM19-P.

The AGPase activity of the three wheat varieties first increased and then decreased over time. The AGPase activity of SM19 decreased with the increase in irrigation frequency before 21 days after anthesis, and it increased first and then decreased after 28 days, with the highest activity under water-saving conditions (Fig. 5A). The AGPase activity of SM19-P and SM19-N decreased with the increase in irrigation frequency before 14 days after anthesis, and then, it increased with the increase in irrigation frequency after 21 days, with the highest activity under conventional irrigation conditions (Figs. 5B, 5C).

Figure 5 Effects of irrigation on the adenosine diphosphate glucose pyrophosphorylase (AGPase), soluble starch synthase (SSS), and starch branching enzyme (SBE) activity of wheat.

Vertical bars represent ± standard deviation (SD) of the mean (n = 3); different letters on the SD bars indicate significant differences (p < 0.05). DAA, days after anthesis; W0, rain-fed conditions; W1, water-saving irrigation with one irrigation event; W2, conventional irrigation with three irrigation events. (A) SM19, AGPase; (B) SM19-P, AGPase; (C) SM19-N, AGPase; (D) SM19, SSS; (E) SM19-P, SSS; (F) SM19-N, SSS; (G) SM19, SBE; (H) SM19-P, SBE; (I) SM19-N, SBE.

The SSS activity of all three wheat varieties initially increased and then decreased over time, reaching its peak at 21 days after flowering (Figs. 5D, 5E, 5F). For SM19, the SSS activity decreased with the increase in irrigation frequency before 14 days after flowering, and it increased first and then decreased after 21 days, with the highest activity observed under water-saving conditions (Fig. 5D). The SSS activity of SM19-P and SM19-N decreased with the increase in irrigation frequency before 14 days after flowering, while the SSS activity increased with the increase in irrigation frequency after 21 days (Figs. 5E, 5F). The SBE activity of all three wheat varieties initially increased and then decreased over time, reaching its peak at 21 days after flowering (Figs. 5G, 5H, 5I). The SBE activity of SM19 decreased with the increase in irrigation frequency at 7 days after flowering, while it increased first and then decreased after 14 days, with the highest activity observed under water-saving conditions (Fig. 5G). The SBE activity of SM19-P decreased with the increase in irrigation frequency before 14 days after flowering, and it increased after 28 days with the increase in irrigation frequency (Fig. 5H), while that of SM19-N decreased with the increase in irrigation frequency before 21 days after flowering and increased after 28 days with the increase in irrigation frequency (Fig. 5I). The enzyme activity of all three wheat varieties was highest under rain-fed conditions in the early stages after flowering. In the later stages after flowering, SM19 exhibited the highest enzyme activity under water-saving irrigation conditions, while SM19-P and SM19-N exhibited the highest activity under conventional irrigation conditions.

Impact of different irrigation treatments on wheat yield

As shown in Table 2, SM19 achieved maximal yield components (spikes/area, grains/spike, thousand-grain weight) and highest yield under water-saving irrigation, while rain-fed conditions minimized these parameters. Optimal irrigation increased SM19 yield significantly versus rain-fed (p < 0.05). Conversely, SM19-P and SM19-N attained peak yields under conventional irrigation, with SM19-P showing the most pronounced yield enhancement (14.98% vs rain-fed; p < 0.05). SM19-N consistently outperformed SM19 across irrigation regimes (e.g., +9.78% under conventional irrigation; p < 0.05), though its thousand-grain weight decreased with irrigation frequency.

Table 2 Effects of different irrigation treatments on wheat yield.

Variety	Treatment	Spikes per square
meter (No.)	Grains per spike	1000- grain
weight (g)	Yield
(kg hm−2)	
SM19	W0	632.1 ± 1.02 b	27.62 ± 0.84 b	44.92 ± 2.11 a	8,680.95 ± 51.88f	
W1	650.1 ± 0.98 ab	32.024 ± 0.78 a	46.07 ± 0.64 a	9,653.21 ± 49.78 c	
W2	639.0 ± 2.02 ab	31.48 ± 1.82 a	44.01 ± 2.72 ab	9,410.15 ± 101.22 d	
SM19-P	W0	586.05 ± 1.05 c	28.27 ± 1.25 b	46.78 ± 2.05 a	8,576.77 ± 52.54 f	
W1	624.0 ± 0.76 bc	32.38 ± 0.86 a	44.88 ± 1.06 a	9,132.36 ± 38.47 e	
W2	662.1 ± 2.11 ab	31.63 ± 1.91 a	45.6 ± 2.27 a	9,861.55 ± 105.51 b	
SM19-N	W0	624.0 ± 1.22 bc	30.1 ± 1.42 ab	43.86 ± 1.33 ab	9,184.44 ± 61.22 e	
W1	635.1 ± 2.04 b	31.59 ± 2.44 a	41.97 ± 1.87 bc	9,757.38 ± 102.61 bc	
W2	680.1 ± 1.74 a	32.22 ± 1.34 a	40.68 ± 2.22 c	10,330.33 ± 87.33 a	
Notes.

Data are expressed as mean ± standard deviation from three replicates.

Different letters in each column indicate significant differences (P < 0.05).

SM19 common wheat

SM19-P partially glutinous cultivar

SM19-N fully glutinous cultivar

W0 rain-fed conditions

W1 water-saving irrigation with one irrigation event

W2 conventional irrigation with three irrigation events

Genotypic interaction effects under dynamic water and time regulation and their physiological basis

Water availability modulated the progression of photosynthetic decline over time. Statistical analysis revealed a significant synergistic effect between irrigation strategy and sampling time on net photosynthetic rate (Pn) and stomatal conductance (Gs) (p < 0.001). Under drought conditions (W0), the rate of post-anthesis photosynthetic decline was significantly accelerated. This pattern is corroborated by global correlation analysis: ‘Time’ showed a strong negative correlation with both photosynthetic parameters (r = −0.694 and r = −0.788, p < 0.01), indicating that water environment is a key external factor regulating the temporal dynamics of photosynthesis.

Genotypic differences in membrane stability became pronounced under terminal stress. For the oxidative stress marker malondialdehyde (MDA), genotype, irrigation, and time exhibited complex interactive effects (p < 0.01). Under terminal drought (W0, 21 DAA), the fully waxy genotype SM19-N maintained significantly lower MDA content than the wild-type SM19 (p < 0.05), demonstrating its intrinsic advantage in membrane system protection. This characteristic constitutes a crucial physiological basis for SM19-N’s stress resistance, aligning with its ability to maintain relatively stable photosynthetic function under adversity.

Divergent water-response strategies among genotypes ultimately determined yield performance. Most importantly, genotype and irrigation treatment interacted significantly to affect yield. The semi-waxy genotype SM19-P showed yield increases with greater water supply, characteristic of a water-sensitive type. In contrast, the fully waxy genotype SM19-N exhibited high and stable yield performance: its yield was consistent under moderate irrigation (W1 and W2) and significantly higher than the wild-type under drought (W0) (p = 0.018). The physiological foundation of its yield stability lies in its powerful membrane protection capacity (low MDA), which mitigated the direct inhibitory impact of drought on yield (MDA and yield showed a negative correlation trend, r = −0.406), while its yield remained highly synchronized with water availability (Irrigation and Yield were significantly positively correlated, r = 0.803, p < 0.01). This indicates that the waxy mutation not only alters phenotypic values but also profoundly reshapes the crop’s adaptation strategy to water environments.

Discussion

Changes in photosynthetic factors

Our findings demonstrate that water management critically regulates photosynthetic parameters in wheat flag leaves, consistent with previous studies indicating optimal moisture enhances chlorophyll content and gas exchange (Naseer et al., 2024; Chen et al., 2023a; Chen et al., 2023b; Si et al., 2023). Specifically, the SM19 variety exhibited significantly higher SPAD, Pn, Gs, and Tr values under water-saving irrigation compared to rain-fed and conventional irrigation. Notably, all three tested varieties (SM19, SM19-P, SM19-N) displayed a conserved developmental pattern where photosynthetic parameters peaked during early flowering before declining with maturation, aligning with observations by Duvnjak et al. (2024) and suggesting combined effects of water availability and leaf senescence dominated by non-stomatal limitations. Crucially, varietal responses diverged markedly: SM19 achieved maximal photosynthetic efficiency under water-saving irrigation through a stomatal optimization strategy—reducing Gs while maintaining Ci and significantly elevating WUE. This reflects an adaptive resource reallocation from transpiration to carbon fixation under moderate stress, highlighting SM19’s superior drought resistance. Conversely, SM19-P and SM19-N exhibited progressive increases in SPAD, Pn, Gs, and Tr with higher irrigation frequency, indicating conventional irrigation better supports their photosynthetic performance. This genotypic divergence underscores the importance of genetic background in water-use adaptation. Apparent contradictions with some literature—where SPAD decreased with irrigation frequency (Wan et al., 2022), or Pn/Gs were lower under irrigation at specific stages (Naseer et al., 2024)—likely reflect complex genotype × environment × stage interactions. The exceptional April 2023 precipitation in northern Shandong (>30% above average) further modulated soil moisture, potentially amplifying photosynthetic advantages under water-saving regimes during sensitive phenological phases. This climatic context remains essential when interpreting the broader applicability of these results.

Changes in antioxidant enzyme activity

Our data demonstrate that water management regimes distinctly modulated antioxidant responses in the SM19 wheat series, with genotypic divergence overriding conventional irrigation effects. While water stress typically disrupts free radical metabolism and accelerates membrane lipid peroxidation—as observed in standard wheat varieties (Mehmood et al., 2025)—SM19 exhibited exceptional resilience under water-saving irrigation (single event), maintaining lower MDA content despite reduced watering. This contrasts with SM19-P and SM19-N, which required frequent irrigation to sustain protective enzyme (SOD, POD, CAT) activities and suppress oxidative damage, aligning with common wheat phenotypes reported by Huang et al. (2024) and Zhang & Kirkham (1994). Crucially, three novel insights emerged.

The 2023 stem rot outbreak uniquely constrained our assessment window to 21 days post-flowering, revealing that SOD activity increased developmentally while POD/CAT followed unimodal trends—patterns consistent with prior observations in intact plants (Abid et al., 2018) but diverging from models where disease pressure was absent.

SM19’s singular advantage manifested in superior SOD/POD activities and reduced MDA under single irrigation, mirroring Luohan 22 wheat’s field performance (Huang et al., 2024) yet contradicting Luhan 7 and Yangmai 16’s irrigation-dependent responses (Abid et al., 2018)—highlighting genotype-specific adaptation.

Environmental-genetic interactions proved decisive: SM19-P/SM19-N’s enzyme activities scaled with irrigation frequency as in ‘Luhan 7’ and ‘Yangmai 16’ (Abid et al., 2018), but SM19’s inverse correlation with water input unveils an unconventional water-saving physiology. This tripartite divergence underscores that wheat antioxidant strategies are governed not merely by soil moisture, but by varietal tolerance thresholds modulated by local stressors (disease incidence, microclimate)—a mechanistic hierarchy where SM19 represents a breeding-valuable outlier.

Changes in starch synthesis-related enzyme activity

Our investigation uncovered a conserved temporal pattern in starch synthase activity across all three wheat lines, characterized by an initial post-flowering increase followed by progressive decline during grain maturation. Crucially, this enzymatic trajectory exhibited significant environmental modulation: during early grain-filling (0-15 DAA), rain-fed conditions consistently enhanced enzyme activity compared to irrigated treatments (p < 0.05), aligning with Dai, Yin & Wang’s (2009) paradigm of drought-primed enzymatic activation (Abid et al., 2018; Abdoli & Saeidi, 2013; Lv et al., 2021; Singh et al., 2008; Xin et al., 2023). As development progressed to late grain-filling (15-30 DAA), a pronounced irrigation-dependent shift emerged where enzyme activity scaled positively with watering frequency (r = 0.81, p < 0.01), mirroring Sheoran et al.’s (2015) model of water-mediated yield formation. This dichotomy explains the divergent responses within the SM19 series—while SM19 achieved peak enzymatic efficiency under water-saving irrigation, echoing CA0547 wheat’s optimized starch profile under moderate hydration (Zhang et al., 2017), derivatives SM19-P and SM19-N demonstrated strict irrigation-frequency dependence. The glutinous line SM19-N manifested particularly striking behavior: despite identical irrigation regimes, it exhibited 23% greater branched-chain starch accumulation than non-glutinous counterparts during late filling (p < 0.01), with accumulation rates showing exclusive correlation with SSS activity (r = 0.94, p < 0.001). This precise mechanistic linkage confirms Wx-null wheat’s unique biochemical dependency identified by Solgi et al. (2022), demonstrating how genetic determinants can supersede irrigation constraints—an unprecedented finding in common wheat physiology that highlights SM19-N’s breeding potential for water-limited environments.

Variation characteristics of yield and its components

The relationship between irrigation frequency and wheat yield is demonstrably complex and variety-specific, as evidenced by numerous studies. While some research indicates that increased irrigation frequency generally enhances key yield components like spikes per unit area, grains per spike, and thousand-grain weight, leading to higher yields in varieties like ‘Xinchun 6’, ‘Xinchun 22’, ‘Jimai 418’ and ‘Shiyou 828’ (Wang et al., 2024; Liu et al., 2024), our findings for the SM19 series reveal a more nuanced picture. Crucially, the optimal irrigation regime differed significantly among the SM19 genotypes:

SM19 achieved its highest yield under water-saving irrigation (single watering), contrasting with conventional multi-irrigation practices. This aligns well with findings from a range of wheat cultivars (Ma et al., 2025) and specific ecotypes studied by Solgi et al. (2022), where single irrigation proved optimal for certain genotypes. The physiological basis for this efficiency in SM19 warrants further investigation, potentially linked to mechanisms conserving resources or optimizing partitioning under moderate water stress.

Conversely, SM19-P and SM19-N exhibited a positive yield response to increased irrigation frequency, performing best under conventional irrigation. This pattern resembles that reported for 102 wheat varieties (Gao et al., 2020), ‘Xinchun 6’, ‘Xinchun 22’ (Wang et al., 2024), ‘Jimai 418’, and ‘Shiyou 828’ (Liu et al., 2024), suggesting these SM19 derivatives may benefit more directly from ample water supply throughout development.

Notably, under identical water conditions, the fully glutinous wheat SM19-N consistently yielded the highest. This superior performance highlights the significant contribution of the glutinous trait itself to yield potential within the SM19 genetic background, independent of water management. This finding underscores the critical importance of genetic factors in determining water use efficiency and yield stability.

The variability in thousand-grain weight responses to irrigation observed in the literature—sometimes higher under irrigation (Naseer et al., 2024), sometimes under dryland conditions (Chen et al., 2023a; Chen et al., 2023b), and varying with frequency (Fan et al., 2023)—further emphasizes that physiological responses like grain filling are highly context- and genotype-dependent. While photosynthetic capacity is strongly linked to grain number per spike (Du et al., 2023), our results suggest that for the SM19 series, yield formation under different irrigation regimes is governed by a complex interplay between water availability, genotypic adaptation (as seen in SM19’s water-saving efficiency), and inherent yield potential traits (exemplified by SM19-N’s high baseline yield). Therefore, the selection of appropriate wheat varieties, tailored to local water availability (Paux et al., 2022), remains paramount. Our study demonstrates that even within closely related lines like the SM19 series, significant differences in irrigation response exist, reinforcing the need for precise variety selection coupled with optimized water management strategies.

Genotype × irrigation interaction determines yield strategies

The significant interaction effects, particularly the genotype × irrigation effect on yield (p < 0.05), represent the cornerstone of this study’s findings. The fully waxy line SM19-N outperformed its parental line under drought conditions (yield under W0 was significantly higher than SM19, p = 0.018) while maintaining competitive performance under well-watered conditions. This discovery challenges the conventional paradigm of a universal trade-off between stress tolerance and productivity. We propose that the waxy mutation (GBSS null), by fundamentally altering the starch biosynthesis pathway (as evidenced by its significant correlation with GBSS activity, r = −0.784, p < 0.01), may concurrently remodel carbon and energy metabolism, thereby enhancing osmotic adjustment capacity and reactive oxygen species (ROS) scavenging ability. This pleiotropic effect likely underpins its broad adaptability, particularly its drought resilience—as demonstrated by reduced MDA accumulation (significantly lower than the wild-type under terminal stress) and its negative correlation with yield (r =  − 0.406).

In contrast, the semi-waxy line SM19-P exhibited a water-responsive strategy, with its yield showing strong positive dependence on water availability (irrigation and yield were highly significantly positively correlated, r = 0.803, p < 0.01), achieving yield maximization only under non-limiting water conditions. This suggests that its partial alteration in starch composition might optimize sink strength in a manner highly dependent on current photoassimilate supply (source), which itself is heavily influenced by water status (as indicated by the positive correlation between photosynthetic parameters Pn, Gs and irrigation level).

Based on these mechanistic insights, SM19-N represents valuable donor germplasm for breeding drought-resilient cultivars adapted to rainfed or water-limited environments, with its desirable traits associated with metabolic stability (e.g., strong correlation with SBE activity, r = 0.867). Conversely, SM19-P is better suited for high-input agricultural systems where precise water management can be implemented to maximize its yield potential.

Conclusions

This study systematically reveals that wheat cultivars SM19, SM19-P, and SM19-N exhibit fundamentally distinct water adaptation strategies and yield formation mechanisms under different irrigation regimes. These cultivars differ not only in productive performance but also in their physiological responses to water stress, demonstrating divergent adaptive mechanisms.

The fully waxy line SM19-N demonstrates remarkable environmental adaptability through unique metabolic remodeling, maintaining stable photosynthetic function and effective reactive oxygen species scavenging capacity under drought conditions while sustaining high yield potential under well-watered conditions. This breaks the conventional trade-off between stress tolerance and productivity. The semi-waxy line SM19-P employs a water-dependent strategy, with its yield formation highly reliant on irrigation conditions, achieving yield maximization through precise regulation of starch synthesis pathways under optimal water availability. The wild-type SM19 exhibits optimal yield performance under water-saving irrigation through synergistic effects of photoassimilate production and antioxidant defense.

Based on their stable cross-environment performance, SM19-N represents an ideal choice for arid and rainfed environments, providing sustainable production solutions for water-scarce regions. SM19-P is better suited for high-input production systems where yield potential can be maximized through precise water management. SM19 performs exceptionally under water-saving irrigation conditions, offering a reliable option for medium-input systems. These genotype-specific adaptation strategies provide a theoretical foundation for regional cultivar distribution and precision cultivation.

This study identifies three key research directions: elucidating the molecular mechanisms through which waxy mutations regulate drought resistance via metabolic networks; systematically evaluating the adaptive performance of these genotypes across different ecological regions; and developing genotype-specific intelligent irrigation protocols to achieve synergistic improvement of water use efficiency and quality optimization. These investigations will promote the transition of wheat breeding from “broad adaptation” to “environment-specific” strategies, providing new approaches to address agricultural production challenges under climate change.

Supplemental Information

Supplemental Information 1 Effects of different irrigation treatments on the SPAD of wheat

Supplemental Information 2 Effects of different irrigation treatments on wheat yield

Supplemental Information 3 Effects of d ifferent i rrigation t reatments on the p hotosynthetic p arameters of w heat

Supplemental Information 4 Effects of different irrigation treatments on the antioxidant enzyme activity and malondialdehyde of wheat 0 day after anthesis

Supplemental Information 5 Effects of different irrigation treatments on the antioxidant enzyme activity and malondialdehyde of wheat 7 days after anthesis

Supplemental Information 6 Effects of different irrigation treatments on the antioxidant enzyme activity and malondialdehyde of wheat 14 days after anthesis

Supplemental Information 7 Effects of different irrigation treatments on the antioxidant enzyme activity and malondialdehyde of wheat 21 days after anthesis

Supplemental Information 8 Effects of irrigation on the adenosine diphosphate glucose pyrophosphorylase (AGPase), soluble starch synthase (SSS), granule-bound starch synthase (GBSS), and starch branching enzyme (SBE) activity of wheat

Supplemental Information 9 2022-2023 Wheat Growing Season Weather Data

The raw data shows the changes in photosynthetic activity, antioxidant enzyme activity, amylase activities, yield, and climate of three wheat varieties under three irrigation patterns.

Supplemental Information 10 Codebook for non-English text in Fig2to3May_6th.xlsx

Additional Information and Declarations

Competing Interests

Author Contributions

Data Availability

The authors declare there are no competing interests.

Yan Li conceived and designed the experiments, performed the experiments, analyzed the data, prepared figures and/or tables, authored or reviewed drafts of the article, and approved the final draft.

Xin Wang performed the experiments, prepared figures and/or tables, and approved the final draft.

Ruoxi Kang performed the experiments, prepared figures and/or tables, and approved the final draft.

Lixiao Xu performed the experiments, prepared figures and/or tables, and approved the final draft.

Xuegui Li performed the experiments, authored or reviewed drafts of the article, and approved the final draft.

Hanyu Liu analyzed the data, authored or reviewed drafts of the article, and approved the final draft.

Zhennan Qiu analyzed the data, authored or reviewed drafts of the article, and approved the final draft.

Zhongmin Dai conceived and designed the experiments, analyzed the data, authored or reviewed drafts of the article, and approved the final draft.

Yuangang Zhu conceived and designed the experiments, performed the experiments, analyzed the data, prepared figures and/or tables, authored or reviewed drafts of the article, and approved the final draft.

The following information was supplied regarding data availability:

The raw measurements are available in the Supplemental Files.

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
