# Peer review of "Dynamic effects of irrigation on photosynthesis and yield-related physiological characteristics in different glutinous wheat cultivars"

_PeerJ, doi:10.7717/peerj.20230_

## Round 0.1 · original submission · Major Revisions

Revise your manuscript based on the reviewers' comments and resubmit it. Also, emphasize the importance of the wheat being fully glutinous, partially glutinous, or non-glutinous in its response to varying irrigation levels. Explain clearly how you determined and correlated this.

**Language Note:** The review process has identified that the English language must be improved. PeerJ can provide language editing services - please contact us at [email protected] for pricing (be sure to provide your manuscript number and title). Alternatively, you should make your own arrangements to improve the language quality and provide details in your response letter. – PeerJ Staff

Reviewer 1 ·

Basic reporting

Generally, it is written in fluent English. It is nice that the study is supported by literature, but sometimes there are many gaps in the sentences. The figures are explanatory, but especially for Figures 2, 3 and 4, different patterns can be selected for WO, W1 and W2, which are used especially when creating the figures. In this case, they are very similar to each other and sometimes it is difficult to distinguish them.

Experimental design

The work is good, but there are some shortcomings in the method. If these deficiencies are corrected, Mamak will be better. for example Was basic fertilization applied in the trial (N, P, K) etc. These are important because yield increase cannot be achieved with irrigation factor alone. If basic fertilization was applied, how much of which fertilizer was given should definitely be added.
-Climate data should be provided from 2022 to 2023. For example, how much rainfall occurred during this period, how many sunny days were there? Since the main theme of the study is irrigation, climate data must be included.
-Why was the photosynthesis measurement done between 9.30-11.30 and spad measurement 12.00? A sentence or two can be added about the importance of taking measurements at these hours

Validity of the findings

The results are generally given as increased or decreased in percentage. However, it is very important to give numerical values ​​for some parameters in order to better understand the results.

Both numerical results and statistical analysis findings should be summarized for all parameters.

Too many percentages and comparisons are lined up one after the other, making the text denser and less readable. If this data is already shown in a table, it should be presented more concisely in the text.

The discussion section is quite weak. Only literature studies on the subject are given in this section. There is not enough discussion about the results obtained, so the discussion should be rewritten.

Additional comments

Dynamic effects of irrigation on photosynthesis and yield-related physiological characteristics in different glutinous wheat cultivars
-The soil analysis result of the area where the experiment is set up must be included in the study.

-Different patterns can be selected for WO, W1 and W2, which are used especially when creating the figures. In this case, they are very similar to each other and sometimes it is difficult to distinguish them.

Was basic fertilization applied in the trial (N, P, K) etc. These are important because yield increase cannot be achieved with irrigation factor alone. If basic fertilization was applied, how much of which fertilizer was given should definitely be added.

-Climate data should be provided from 2022 to 2023. For example, how much rainfall occurred during this period, how many sunny days were there? Since the main theme of the study is irrigation, climate data must be included.

-Why was the photosynthesis measurement done between 9.30-11.30 and spad measurement 12.00? A sentence or two can be added about the importance of taking measurements at these hours

RESULTS
Line 155-179 In sentences where percentage changes are stated, control values are not given. This makes it difficult for the reader to better assess the magnitude of the increase rates.

Both numerical results and statistical analysis findings should be summarized for all parameters.

Line 180-227 In the conclusion section, enzyme activity values of SM-19 should be given in more detail. Numerical values should be given rather than % data so that the comparison made with SM19-P and SM19-N can be clearly understood.

Too many percentages and comparisons are lined up one after the other, making the text denser and less readable. If this data is already shown in a table, it should be presented more concisely in the text.

Discussion
When making direct comparisons between sources, differences in species, environmental conditions or irrigation regimes are not sufficiently emphasized. If there are differences (e.g. irrigation time, soil type, climate), this should be highlighted in this context.

The “new” contribution of the current study is not clearly emphasized. It should be clarified what the new statement of this study is, rather than simply repeating similar findings in the literature.

Recommendation: “Unlike previous studies focusing solely on X or Y varieties, our study highlights the response of partially and fully glutinous wheat varieties (SM19-P, SM19-N) under varying irrigation frequencies.”

The physiological mechanism of the findings should be explained more clearly. For example:
Why does single irrigation result in the highest Pn in some varieties?

How permanent is the effect of irrigation frequency on chlorophyll content?

Line 334 “Compared with no irrigation, the frequency of irrigation had no effect on the changes in the Pn and Gs of Jinmai 47...”

The “no effect” result here contradicts other findings; brief comments should be made on the reasons for these contradictions: cultivar differences, experimental conditions, environmental effects?

Line 347 “This suggests that SM19 has stronger drought resistance and can grow better under moderately water-deficient conditions.”

This statement is too general. Is the “drought resistance” statement based on physiological data, yield, or SPAD/Pn values? It should be clarified.

“Photosynthetic efficiency was higher under water-saving conditions.” should be stated more clearly on what is considered ‘higher’ (e.g. SPAD, Pn?).

Although the sources are very well integrated, the transitions between sentences are weak. For example:
Line 419 "The synthesis of branched-chain starch occurs earlier than straight-chain starch synthesis..." this sentence comes in suddenly and is not connected to the previous sentence.

Some important statements are left too general and lack supporting explanations:
Line 430 "Irrigation is beneficial to the synthesis of starch in wheat grains during the late grain-filling period."

What type of irrigation? At what level? In which varieties? Such statements need more detail.
The responses of enzyme activities of varieties SM19, SM19-P and SM19-N to irrigation regimes should be explained in more detail. Statements such as "increased with frequency" alone are not sufficiently explanatory.

Paragraphs are generally very long and fragmented, making it difficult for the reader to follow the main points.

Some statements are connected to each other in contradictory ways:

Line 457-461 The relationship between irrigation frequency and wheat yield is not necessarily positively correlated. Previous research with 18 different wheat ecotypes found that irrigation significantly increased the number of spikes and the yield of wheat, but there was no significant difference between one irrigation event and two irrigation events.

Conclusions
Some sentences are too long, complicated, or contain ambiguous expressions:
“...various factors are coordinated, achieving the highest yield.”
→ Which factors? Unclear. Should be more specific.

The following sentence is grammatically incomplete:
“SM19-N wheat is suitable for areas with different moisture conditions, high photosynthetic efficiency, SOD, POD, and SBE activities play an important role.”

→ Two independent sentences are connected by a punctuation error. Such expressions violate the rules of scientific writing.

Results Should Be More Concrete:

For the results to be more effective, it is recommended that they be supported by quantitative data such as yield increase (e.g., “yield increased by X%”).

Instead of general expressions such as “Varieties respond differently,” which variety is more advantageous under which conditions and in terms of which physiological trait should be presented more specifically.

Application Recommendations Should Be Clearer:

The last bullet point says that SM19-N is "suitable for areas with different moisture conditions", but it does not detail how successful it is under which conditions.

More direct recommendations for wheat producers could be added: for example, "In moderately irrigated zones, SM19-N is preferred due to..."

Reviewer 2 ·

Basic reporting

This is a very original and beautifully designed work. However, the one-year termination of the study shows that it cannot have scientifically supportive data. The abstract does not reflect the overall integrity of the study. Not including climate data in the study will cause deficiencies in the interpretation of the results obtained. Especially the material, methodology and findings section is very simple and incomplete. Detailed method could not be given. As important as the data obtained are, explaining them constitutes another part of the article. The academic language of the study is very weak. For these and similar reasons, the study is not publishable.

Experimental design

106- The plot size seems quite small.
114- Pn? It must be clearly stated. Why did you receive this data? When did you get it? How did you get it? It should have been clear and detailed.
115- No detailed information and reference about the method.
152- I think the information on data analysis should not be limited to this.

Validity of the findings

174- Why is there a very serious drop in SPAD and Pn in the 14th DAA? This is not described. And it is stated to be in the same direction as the other parameters. This needs to be checked.
298 - Why does the SM19 have a different response at W2? These and similar situations are not explained in detail.

Additional comments

Although the study is a labor-intensive work, it is very poor in terms of writing. The issues addressed in the discussion section are not consistent with the integrity of the study. At the same time, the shorter the findings section, the longer the discussion section.

Reviewer 3 ·

Basic reporting

The manuscript is generally written in clear and professional English; however, several grammatical and stylistic issues require revision for improved clarity and flow. The abstract includes key findings but suffers from awkward phrasing and ambiguous syntax. The final sentence of the abstract (lines 31–32) lacks grammatical coherence and contains overlapping ideas that need to be separated or rewritten for clarity. I suggest splitting the sentence into two and ensuring subject–verb agreement.

The manuscript structure adheres to PeerJ standards. The introduction appropriately situates the study within the context of water stress in wheat cultivation, citing relevant and recent literature. While previous studies are discussed, the novelty of applying irrigation strategies specifically to glutinous wheat could be emphasized more explicitly.

The figures are numerous and generally well-constructed. However, they are very dense-particularly Figures 1, and 5. Consider combining or simplifying where appropriate to improve visual digestibility.

Experimental design

The experimental design is scientifically sound, and the methodology is largely well described. The study involves three irrigation treatments applied to three genetically related wheat cultivars, offering a robust factorial design. The authors conducted the experiment in randomized complete blocks with three replicates, which is appropriate for field studies of this nature.

The selection of glutinous and non-glutinous wheat cultivars is justified, but more explanation is needed regarding why these three lines were chosen and whether their drought tolerance and starch traits were known prior to the experiment. The genetic background of the glutinous lines (SM19-P and SM19-N) is briefly described in lines 97–101; however, more information on their agronomic relevance and prior physiological behavior under drought would enhance the reader's understanding.

The environmental context of the field trial, especially climatic conditions (temperature, rainfall), is not fully described. As water availability is central to this study, the authors should provide detailed meteorological data, either in the text or as a supplementary figure. While irrigation volumes are clearly outlined in Table 1, without rainfall information, it is difficult to assess the total soil water balance experienced by each treatment.

Validity of the findings

The findings are robust, and the statistical analysis appears to be appropriate. The use of repeated measurements across multiple developmental stages enhances the temporal resolution of the data and allows for a more comprehensive evaluation of physiological responses.

The conclusions drawn from the data are generally supported by the results. The differential responses of SM19 versus SM19-N to irrigation regimes are particularly well articulated. However, while the authors show that irrigation significantly influences enzyme activity and physiological performance, they do not statistically test correlations between physiological traits and yield components. Including a correlation matrix or regression analysis would strengthen the causal interpretations made-especially the suggestion that higher enzyme activities or SPAD values lead to higher yields.

Additionally, some of the yield improvements reported (e.g., 6.24% to 14.98%) are relatively modest. While statistically significant, the practical significance of these gains should be discussed more thoroughly, especially in the context of water use efficiency and irrigation costs.

The discussion is quite detailed and cites a substantial body of literature. Nonetheless, the section could benefit from greater synthesis rather than listing comparisons with previous studies one by one. For example, rather than stating that “A found X, B found Y, C found Z,” the authors might consolidate by saying, “Previous research collectively shows that...,” and then discuss deviations.

---

## Round 0.2 · Minor Revisions

Your manuscript still needs some revision. The result section should be rewritten. It is very confusing. Not present some data in conclusion section. Write your outcome short and give some advice future research. The importance of glutinous wheat and its response to irrigation are still not explained. Your comments are not enough. Why do you need glutinous? What does glutinous wheat provide tolerance to drought? I think that wheat without glutinous is the appropriate nonirrigated condition. So why would you want to make it glutinous? Please explain

Reviewer 1 ·

Basic reporting

The revised article is written in very clear and fluent language. This new version is much better than the previous one.

Experimental design

All the corrections I suggested for the Materials and Methods section have been made and incorporated into the article. In this respect, a very successful section has emerged.

Validity of the findings

Rewriting the results and discussion sections and supporting them with new literature has significantly improved the article.

Additional comments

I'd like to thank you, the authors, for explaining each and every one of my suggested corrections. A very clear response and track file have been prepared.

Reviewer 2 ·

Basic reporting

Although I had a different opinion before, the current version is very explanatory and a lot of work has been put into it. For this reason, when I examine the work, I find answers to all my questions and I am convinced that the work is publishable.

Experimental design

.

Validity of the findings

.

---

## Round 0.3 · Minor Revisions

Thank you for your revisions to your manuscript titled "Dynamic effects of irrigation on photosynthesis and yield-related physiological characteristics in different glutinous wheat cultivars." However, there are still some issues I'm not satisfied with. One-way ANOVA is not an appropriate model for data analysis. There are three different genotypes, three irrigation regimes, and different sampling times. However, not everything is fully clarified in the materials and methods section. This was overlooked in our previous reports. Reanalyze your data using a factorial design. Furthermore, I believe the wording of the Conclusion section is still inappropriate. Except for the last paragraph, your entries can be included in the Results section. Please rewrite them based on other previously published articles.

---

## Round 0.4 · Minor Revisions

Dear authors,

Your manuscript is now close to acceptance. However, upon close inspection, we notice that the authors have cited almost exclusively Chinese authors (~97%) whereas a huge volume of literature exists from around the world on wheat growth, yield, physiology, and the effect of irrigation.

Please expand your Introduction to include a more representative sample of the literature on this cosmopolitan crop.

---

## Round 0.5 · accepted · Accept

Your manuscript can be accepted after the revisions.

For example:

Abstract - the acronym DPA should be spelled out on first use. That acronym is not used elsewhere in the manuscript. References -- scientific names need to be written in italics; genus names should be capitalized.